# Waveform distortion for temperature compensation and synchronization in circadian rhythms: An approach based on the renormalization group method

**Shingo Gibo**[iD][1¤*], **Teiji Kunihiro**[2], **Tetsuo Hatsuda**[1,3], **Gen Kurosawa**[iD][1*]

**1** RIKEN Center for Interdisciplinary Theoretical and Mathematical Sciences (iTHEMS), Wako, Japan,
**2** Yukawa Institute for Theoretical Physics (YITP), Kyoto University, Kyoto, Japan, **3** Kavli Institute for the Physics and Mathematics of the Universe (Kavli IPMU), WPI, University of Tokyo, Kashiwa, Japan

¤ Current address: Biomedical Mathematics Group, Pioneer Research Center for Mathematical and Computational Sciences, Institute for Basic Science, Daejeon, Republic of Korea
* shingogibo@ibs.re.kr (SG); g.kurosawa@riken.jp (GK)

**Data availability statement:** All code are available at https://github.com/gibo3/circadian_renormalization. Experimental

## Abstract

Numerous biological processes accelerate as temperatures increase, but the period of circadian rhythms remains constant, known as temperature compensation, while synchronizing with the 24h light-dark cycle. We theoretically explore the possible relevance of waveform distortions in circadian gene-protein dynamics to the temperature compensation and synchronization. Our analysis of the Goodwin model provides a coherent explanation of most of temperature compensation hypotheses. Using the renormalization group method, we analytically demonstrate that the decreasing phase of circadian protein oscillations should lengthen with increasing temperature, leading to waveform distortions to maintain a stable period. This waveform-period correlation also occurs in other oscillators like Lotka-Volterra, van der Pol models, and a realistic model for mammalian circadian rhythms. A reanalysis of known data nicely confirms our findings on waveform distortion and its impact on synchronization range. Thus we conclude that circadian rhythm waveforms are fundamental to both temperature compensation and synchronization.

## Author summary

Our daily rhythms are underlain by gene regulatory and biochemical networks, called circadian clocks. Although most biochemical reactions accelerate as temperature increases, the period of circadian rhythms is almost constant even with increasing temperature. This phenomenon is called temperature compensation, and the mechanism is still unclear. By applying a method of theoretical physics, the renormalization group method to a biological problem, we revealed that the waveform of gene dynamics should be more distorted from sinusoidal wave at higher temperature when the circadian period

data were previously published in Zhou et al. (2015) and Kidd et al. (2015).

**Funding:** This work was supported by grants from the Japan Science and Technology Agency (JPMJCR1913 to G.K.), and from the Japanese Society for the Promotion of Science, and the Ministry of Education, Culture, Sports, Science, and Technology in Japan (21K06105 and 24H02025 to G.K., 19K03872 and 24K07049 to T.K.). The funders had no role in study design, data collection and analysis, decision to publish, or preparation of the manuscript.

**Competing interests:** The authors have declared that no competing interests exist.

is stable to changes in temperature. This prediction as for the importance of waveform in temperature compensation is verified by analyzing published experimental data of Drosophila and mice. Notably, the correlation between period and waveform distortion holds for other oscillator models, indicating the waveform distortion is important for determining the period in various types of oscillatory systems. Another important challenge in understanding circadian clocks is how they synchronize with environmental light-dark cycles. By theoretically analyzing a circadian clock model, we found that the frequency range for synchronization becomes narrower when the waveform is distorted.

## 1. Introduction

Humans exhibit sleep-wake cycles with an approximate 24h period, and these cycles persist under constant environmental conditions, a phenomenon termed the circadian rhythm. This temporal regulation exists in both humans and various organisms such as molds, plants, and insects [1–3]. Recent advances in genetic research through insects, molds, mammals, and plants have unveiled that genes and proteins are involved as integral components in the primary mechanism governing autonomous circadian rhythms [1,4–6].

Understanding circadian rhythms holds promise for deciphering a multitude of sleep patterns, including sleep disorders such as advanced sleep phase syndrome (characterized by early awakening around 4:00 am), delayed sleep phase syndrome (marked by late awakening), non-24h sleep-wake disorder, and narcolepsy [7]. Notably, advanced and delayed sleep phase syndromes are believed to be linked to the circadian rhythm period [8–13]. Ongoing studies explore possible correlations between genetic characteristics revealed by large-scale genetic analysis and various sleep patterns [14–16]. However, the nature of the system is so intricate that it remains a challenge to link sleep patterns to specific genes. In such a situation, it would be meaningful to have recourse to mathematical models and obtain possible hints for the linkage and hopefully suggestions for studies of genetic dynamics.

One fundamental issue that remains to be understood in circadian rhythm research is temperature compensation [17–25], in which the period keeps constant despite temperature-induced changes in reaction rates. Temperature compensation of the period occurs not only in circadian rhythms but also in ultradian rhythms, such as yeast metabolic cycles [22,26]. Despite the extensive experimental and theoretical research on temperature compensation, the mechanism has remained elusive. Hypotheses have been proposed to explain temperature compensation, including the balance hypothesis, critical-reaction hypothesis, temperature-amplitude coupling hypothesis, and waveform hypothesis. The balance hypothesis proposes that the stability of the circadian period with temperature arises from a balance between period-lengthening and period-shortening reactions [17,18,27]. The critical-reaction hypothesis assumes that there should be critical reactions that determine the circadian period. If these reaction rates are stable against temperature variations, then the circadian period will similarly remain stable [28–30]. The temperature-amplitude coupling hypothesis suggests that temperature-sensitive amplitudes in gene activity rhythms should generate a stable period by generating larger amplitudes at higher temperatures [22,31]. Lastly, the waveform hypothesis proposes that temperature-sensitive waveforms in gene activity rhythms should be correlated with a stable period in a manner that their higher harmonic components become larger and the distortion of the waveform increases at higher temperatures [32]. These proposed mechanisms, along with other mechanisms discussed in [33], are not necessarily mutually exclusive and may work in combination to achieve temperature compensation.

Another key issue in circadian rhythm research is synchronization with 24-hour environmental light-dark cycles. Previous theoretical and experimental studies on synchronization revealed that if the internal period of the oscillation closely matches the external period, then it is more likely to synchronize with the forced period [34–37]. Additionally, experimental studies on several species uncovered genes and proteins in circadian systems affected by a light pulse [38,39]. In reality, the circadian rhythm must adjust to the 24h light-dark cycle while maintaining a temperature-compensated circadian period. Therefore, multiple questions arise. (i) Given the significant temperature variations between seasons, how do organisms synchronize their circadian rhythms with the 24h light-dark cycle across various temperatures [20,36,40]? (ii) if the gene activity rhythm of the circadian rhythms becomes more distorted as temperatures increase to achieve temperature compensation, how does the ease of synchronization change with temperature variations? Theoretical analyses incorporating the findings of light pulse experiments might provide further insights into these questions.

In the present paper, we investigate possible roles of the waveform distortion in temperature compensation based on analytical and numerical analyses of the Goodwin model for circadian rhythms and clarify how the waveform in gene activity rhythms tends to be more distorted at higher temperatures (e.g., steeper rise, longer tail) for temperature compensation.

To this end, we employ the renormalization group (RG) method [41–46] adapted for global and asymptotic analysis of differential equations on the basis of the perturbation theory [47–53]. As elucidated in a historical review by Shirkov [54], there are several distinct approaches that have been referred to collectively as the renormalization group (RG) method. Among these approaches, concepts such as asymptotic functional self-similarity, reminiscent of RG techniques used in extracting critical exponents in statistical physics [44–46], were notably applied by Feigenbaum [55] to derive universal constants characterizing bifurcation points in certain iterative maps encountered in population biology [56,57]. In contrast, the RG method utilized in this current work [47,48,51] is an adaptation of the approach originally developed in quantum field theory [42,43]. This approach provides a straightforward resummation technique for perturbative solutions, thus offering practical approximate solutions valid over global time domain. The method also serves as a powerful reduction theory of the dynamics in the asymptotic region. The method has been applied to various models, including ordinary and partial differential equations, discrete maps, and stochastic equations [48–50,52,53,58]. The origin of the powerfulness of the RG method as a tool of the global analysis can be intuitively understood by reformulating the method in terms of the classical theory of envelopes [48,53]: The envelope of a set of perturbative solutions which are only valid only locally around the arbitrary initial time $t_0$ can be set up to constitute a valid solution to the equation in a global domain.

Combining an index for waveform distortion, namely non-sinusoidal power (*NS*) introduced by two of the present authors (KG) [32], with the result of the RG method, we can obtain both a unified picture of the above mentioned theoretical hypotheses (balance hypothesis, critical-reaction hypothesis, temperature-amplitude coupling hypothesis, and waveform hypothesis) and quantify previous experimental data on Drosophila [59]. Our analyses demonstrate that the fundamental role of the waveform distortions in temperature compensation from both theoretical and experimental perspectives in accordance with the previous findings [32]. Moreover, we reveal for the first time the mechanism by which the synchronization of circadian rhythms changes with temperature if the waveform in gene activity rhythms is more distorted at higher temperatures. We theoretically prove that the frequency range of the external force that synchronizes circadian rhythms becomes narrower if the waveform of gene activity rhythms is more distorted. This indicates that it is more

difficult to synchronize with light-dark cycles at higher temperatures. The present result of synchronization is consistent with the previous experimental and numerical studies demonstrating that the magnitude of the phase shift caused by light pulses was smaller at higher temperature [32,60,61].

This paper is organized as follows. In Sect 2, we introduce the index for waveform distortion and the Goodwin model, and summarize the main results regarding temperature compensation and synchronization of circadian rhythms. In Sect 3, we investigate the conditions for temperature compensation. We present our results based on the renormalization group (RG) method in Sects 3.1, 3.2 and 3.3, followed by experimental verification of the model's predictions in Sect 3.4. In Sect 4, we examine the conditions for synchronization. Sect 5, the final section, is devoted to discussions. Some detailed technical account of our method of analysis and numerical analyses are given in S1 Text–S5 Text.

## Results

## 2. Waveform distortion in circadian rhythms

### 2.1. Index for waveform distortion

Let the time dependence of a certain variable in the circadian rhythm system be expressed in a Fourier series as $x(t) = \sum_{j=-\infty}^{\infty} a_j \exp(i(\frac{2\pi}{\tau})jt))$, with $a_j$ being the Fourier coefficients of the oscillatory time series. Then, we introduce an index for describing the distortion of $x(t)$ from a sinusoidal shape as

$$NS = \left[ \frac{\sum_{j=1}^{\infty} |a_j|^2 j^m}{\sum_{j=1}^{\infty} |a_j|^2 j^q} \right]^{\frac{1}{2}} \quad (m > q \geq 0), \tag{1}$$

where $m$ and $q$ are integers. Termed the "non-sinusoidal power ($NS$)", this index is designed to emphasize higher harmonics ($m > q$) as discussed in a previous paper [32]. For instance, we have $NS = 1$ when the time series has a sinusoidal waveform in which only the coefficients for the fundamental component are non-zero ($a_{\pm 1} \neq 0$). Conversely, for non-sinusoidal time series, the coefficients for higher harmonics are non-vanishing, resulting in $NS > 1$. The previous theoretical work demonstrated that a more distorted waveform (larger $NS$) at higher temperature is necessary for temperature compensation in the four-variable negative-feedback model [32]. However, it is unclear whether the relevance of waveforms to temperature compensation found in previous research has general validity not restricted to some specific model.

To explore the possible relevance of the waveform characteristics to temperature compensation and synchronization, we consider the simplest model for circadian rhythms, known as the Goodwin model (Fig 1A) [62]. This model incorporates negative-feedback regulation of gene expression, a mechanism established as essential for transcriptional-translational oscillations. The three-component Goodwin model reads:

$$\frac{dx_1}{dt} = f(x_3) - k_1 x_1, \tag{2}$$

$$\frac{dx_2}{dt} = p_1 x_1 - k_2 x_2, \tag{3}$$

$$\frac{dx_3}{dt} = p_2 x_2 - k_3 x_3, \tag{4}$$

where $x_1(t)$ represents mRNA abundance, and $x_2(t)$ and $x_3(t)$ denote protein abundance. The function $f(x_3)$ in the model signifies transcriptional regulation, and the parameters $p_1$

and $p_2$ denote protein synthesis and phosphorylation rates, respectively, and $k_i$ ($i$ = 1, 2, 3) represent degradation rates (Fig 1A). In fact, there are two versions of the Goodwin model: (i) one in which the degradation rate is expressed as a linear function of the substrate, as in Eqs (2)–(4), and which requires high cooperativity ($n$) for oscillations; and (ii) another formulation in which the degradation rate follows a Michaelis-Menten equation, where a cooperativity of one is sufficient to generate oscillations [63–65]. In the present paper, we assume that the degradation rate follows a linear function of the substrate, for the sake of simplicity and analytical tractability. Forger derived the period of Goodwin model with linear degradation (2)–(4) by applying signal processing methods [66], while a similar analysis has also been conducted for other negative feedback oscillators [67]. First, Eqs (2)–(4) were transformed into a single equation with higher-order derivatives (see Eq (47) in S2 Text). Next, the nonlinear term $f(x_3)$ was eliminated by multiplying the transformed equation by $dx_3/dt$ and integrating for one period. Then, by expressing the periodic solution in Fourier series as $x_3(t) = \sum_{j=-\infty}^{\infty} a_j \exp(i(\frac{2\pi}{\tau})jt))$, the period was derived as follows:

$$\tau = \frac{2\pi}{\sqrt{k_1 k_2 + k_2 k_3 + k_3 k_1}} \left[ \frac{\sum_{j=1}^{\infty} |a_j|^2 j^4}{\sum_{j=1}^{\infty} |a_j|^2 j^2} \right]^{\frac{1}{2}}, \tag{5}$$

where $a_j$ is the Fourier coefficient of $x_3(t)$. If $x_3(t)$ is distorted from a sinusoidal wave, the higher-order components $a_j$ ($j \geq 2$) become large, causing $NS$ to be large. Subsequently, two of the present authors showed that this formula implies that temperature compensation of the period in this model occurs only when the waveform ($NS$) is distorted as temperature increases [32]. Suppose that all reactions become faster as temperature increases in the model. Then, one can numerically demonstrate that the waveform tends to be more distorted (larger $NS$) at higher temperatures for temperature compensation (Fig 1B, magenta line). We call this mechanism the waveform hypothesis for temperature compensation. This result, based

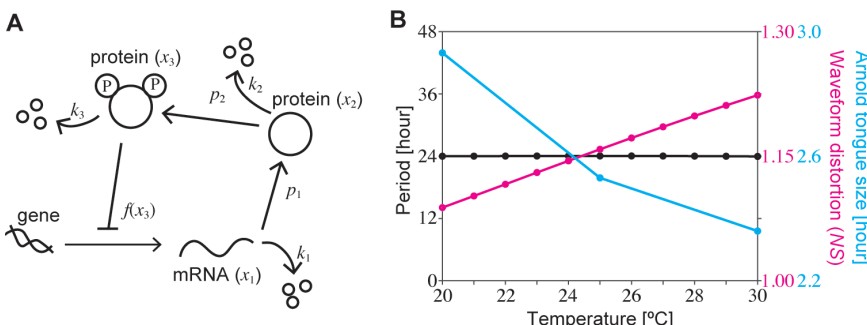

**Fig 1. (A) The circadian clock model and (B) the relevance of waveform characteristics to temperature compensation (*magenta*) and synchronization (*cyan*).** The transcriptional-translational function is defined as $f(x_3) = r/(1 + (x_3/K)^n)$, with the values of $K$ and $n$ set to be 0.0184 and 20, respectively. Assuming that the parameters $k_1$, $k_2$, $k_3$, $p_1$, $p_2$, and $r$ increase with increasing temperature, we model the reactions rates using the Arrhenius law $b_i = A_i \exp(-E_i/RT)$, where $b_i$, $E_i$, $A_i$, $R$, and $T$ are the rate constants, activation energies, frequency factors, gas constant ($R$ = 8.314), and absolute temperature, respectively. The values of $E_i$ and $A_i$ are detailed in S1 Table. The synchronization region depends on the period of light-dark cycles and light intensity, known as the Arnold tongue [37]. The size the Arnold tongue is defined by the difference between the lower limit of synchronization, $2\pi/\Omega_{\text{large}}$, and the upper limit of synchronization, $2\pi/\Omega_{\text{small}}$. When the period is constant in a different temperature, the waveform distortion ($NS$) becomes larger, and the synchronization range (size of the Arnold tongue) becomes narrower with increasing temperature as shown in (B) in the case of the light intensity, $I$ = 0.001.

on the Goodwin model [32], which demonstrated that a more distorted waveform at higher temperatures is both necessary and sufficient for temperature compensation, aligns with a previous study [67] showing that an alternative oscillator model with sinusoidal oscillations alone cannot achieve temperature compensation.

## 2.2. Theories of temperature compensation

The waveform hypothesis and the other three hypotheses for temperature compensation (balance hypothesis I, critical-reaction hypothesis II, and amplitude hypothesis III) are not mutually exclusive, which can be understood in a unified way through Eq (5):

I.   The balance hypothesis, previously explored theoretically by Ruoff [18], suggests that the temperature compensation of the period is caused by a balance between the effects of reactions that shorten the period and those that lengthen the period. Eq (5) illustrates that the balance between the effect of shortening the period and that of lengthening the period can be caused by changing the distortion of the waveform.

II.   Eq (5) demonstrates that even if some of the governing reactions in circadian rhythms are temperature-insensitive [28,30] as discussed in the critical-reaction hypothesis. Still, the temperature compensation requires waveform distortion at high temperatures if reactions other than some of the governing reactions accelerate at higher temperatures.

III.   Our numerical analysis of the circadian model (Fig 1B) show that when temperature compensation occurs, the waveform is more distorted at higher temperatures, and the amplitude of the oscillation is larger (S1 Fig). This tendency for the amplitude to increase at high temperatures is consistent with the amplitude hypothesis. According to Eq (5), a greater distortion of the waveform at higher temperatures is necessary, but not sufficient, for temperature compensation.

## 2.3. Synchronization in circadian rhythms

If the waveform is more distorted at higher temperatures for temperature compensation, then it would be intriguing to explore whether the temperature- dependent waveform also affects the synchronization of circadian rhythms with environmental light-dark cycles at various temperatures. Theoretical and experimental studies of synchronization and circadian rhythms illustrated that the oscillation is more likely to synchronize with forcing cycles if the internal period is sufficiently close to the period of the forcing cycles [34–37]. In mammals and Neurospora, a light pulse is known to increase Per1 and frq mRNA expression [38,39]. To incorporate gene activation during the light phase, we employ a model in which Eq (2) for the change in mRNA expression is modified to

$$\frac{dx_1}{dt} = f(x_3) - k_1 x_1 + I\cos(\Omega t),\qquad(6)$$

where $I$ represents the light intensity and $\Omega$ is the angular frequency of the light-dark cycles.

## 2.4. Main results

In Sects 3 and 4, we provide detailed discussions on how the waveform in gene activity rhythms should be distorted at higher temperatures using the RG method, as well as the synchronization in circadian rhythms. The main results are pictorially summarized in Fig 1B. The magenta line indicates that the waveform of the gene activity rhythms should be more distorted at higher temperatures for temperature compensation, whereas the cyan line

indicates that synchronization with the light-dark cycles should become more difficult at higher temperatures because of the larger waveform distortion.

## 3. Waveform distortion and temperature compensation

### 3.1. Numerical simulation of the waveform-period correlation

Eq (5) and numerical simulations indicate that $NS$ tends to be larger when the period is relatively stable even with increased parameter values (see Fig 1B). To quantitatively reveal the correlation between waveform and period, we conduct numerical simulations using a circadian clock model. In the analysis of the circadian clock model, the transcription function $f(x_3) = r/x_3^n$ was considered for simplicity. We first search for parameter sets in which oscillations occur. We define those parameter sets as the reference parameter sets. Because many biochemical parameters have not yet been measured, we prepared 100 random reference parameter sets for the oscillations. $k_1$, $k_2$, $k_3$, $p_1$, $p_2$, and $r$ were assigned uniformly distributed random values ranging from 0 to 10, and $n$ was assigned a uniformly distributed random integer ranging from 9 to 15. The period obtained with each reference parameter set was denoted as $\tau_1$. Next, reaction rates often follow the Arrhenius equation, which states that a 10°C rise in temperature increases the reaction rate by a factor of 2-3. To incorporate the effect of high temperature, instead of using the Arrhenius equation, each parameter in the model's reference parameter set was randomly multiplied by a factor of 1.1-1.9, and the period and waveform were examined when the oscillation behavior persists. The period obtained by increasing the parameters from each reference parameter set was denoted as $\tau_2$, and the ratio, $\tau_2/\tau_1$, was called the relative period.

To quantitatively analyze the correlation between period and waveform when temperature compensation occurs, we consider the case of the relative period $\geq 0.85$ because it has been experimentally confirmed that the circadian rhythm frequency at high temperature divided by that at low temperature of the wild-type ranges between 0.85 and 1.15 when the temperature is increased by 10°C and temperature compensation occurs [19]. In the present numerical analysis, the period is relatively stable (relative period $\geq 0.85$) in 34 of the 4900 parameter sets, in qualitative agreement with previous theoretical analyses that the period often shortens with increasing reaction rates [18,27]. Because the range of parameter variation is 1.1-1.9 and the average value is 1.5, the reaction rate is accelerated by a factor of 1.5 on average, and the average relative period is approximately $1/1.5 \approx 0.67$. Fig 2 indicates that when temperature compensation occurs, there is a clear correlation between the period and waveform.

### 3.2. Brief introduction of the renormalization group method

In this subsection, we give a brief introduction to the renormalization group method for obtaining a useful approximate solution that is valid in a global domain from the perturbative solution in a way presented in [48,49,52].

Given a differential equation with $t$ being the independent variable, naive perturbative solutions to it generically contain secular terms proportional to polynomials of $t$, such as $t^n \cos(\omega t)$ or $t^n \sin(\omega t)$ with $n$ being an integer when the unperturbed solution is given by a harmonic oscillation with an angular velocity $\omega$. Thus the naive perturbative solution would show a divergent behavior and goes far away from the exact solution for large $t$ because of the secular terms. The RG method gives a simple resummation method of the perturbation series and hence a globally valid solution utilizing the initial condition with the integral constants at arbitrary time $t_0$.

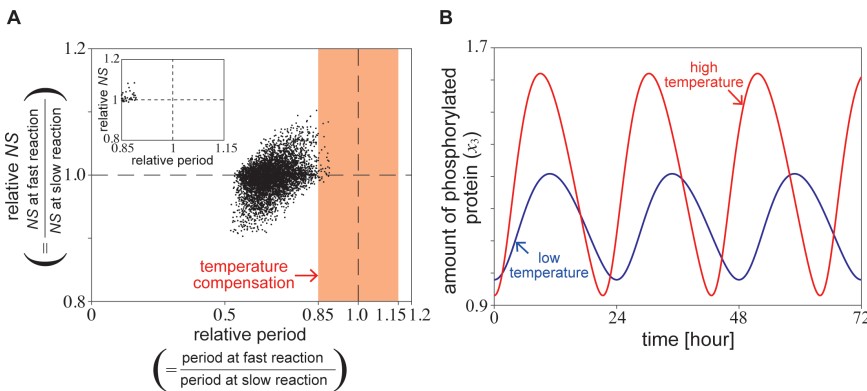

**Fig 2. (A) Distribution of relative** $NS$ **for** $x_3(t)$ **of the circadian clock model as a function of the relative period when reaction rates are increased.** We first generated reference parameter sets. To incorporate the effect of the increase of temperature, instead of using the Arrhenius equation, each parameter $k_1$, $k_2$, $k_3$, $p_1$, $p_2$, and $r$ in the model's reference parameter set was randomly multiplied by a factor of 1.1-1.9. The insertion indicates the region of temperature compensation ($0.85 \leq$ relative period $\leq 1.15$). (B) Examples of the waveform in Goodwin model when the period was relatively unchanged. The blue and red lines represent slow and fast reactions, respectively.

To show this, let us take the following linear damped oscillator as a simplest example [48]:

$$\frac{d^2x(t)}{dt^2} + 2\epsilon\frac{dx(t)}{dt} + x(t) = 0 \tag{7}$$

with $0 < \epsilon < 1$. The exact solution is given by

$$x(t) = A(t)\cos\phi(t) \tag{8}$$

where $A(t) = A_0\exp(-\varepsilon t)$ and $\phi(t) = \omega t + \phi_0$ with $\omega = \sqrt{1 - \varepsilon^2}$: Owing to the velocity-dependent friction $\epsilon dx(t)/dt$, the amplitude has a time-dependence and the angular velocity is shifted from the unperturbed value 1; note that the solution (Eq (8)) can be expressed as an infinite power series of $\epsilon$ although the $\epsilon$ enters the Eq (7) linearly.

Now we dare to apply the perturbation theory to get a solution $x(t; t_0)$ of the linear equation (Eq (7)) with the initial value $W(t_0)$ at an *arbitrary* time $t = t_0$,

$$x(t = t_0; t_0) = W(t_0),$$

which is unknown but yet to be determined by the following procedure. The perturbation solution $x(t; t_0)$ is expressed as a power series of $\epsilon$ as $x(t; t_0) = x_0(t; t_0) + \epsilon x_1(t; t_0) + \epsilon^2 x_2(t; t_0) + o(\epsilon^2)$. In the RG method, we assume that the initial value $W(t_0)$ is on an exact solution, and is also expressed as a power series of $\epsilon$ as $W(t_0) = W_0(t_0) + \epsilon W_1(t_0) + \epsilon^2 W_2(t_0) + o(\epsilon^2)$ with $x_i(t_0; t_0) = W_i(t_0)$ ($i = 0, 1, \dots$).

The zeroth-order equation reads

$$\frac{d^2x_0}{dt^2} + x_0 = 0,$$

the solution to which may be expressed as $x_0(t; t_0) = A(t_0)\cos(t + \theta(t_0))$ with $W_0(t_0) = A(t_0)\cos(t_0 + \theta(t_0))$. Here we have made explicit the fact that the amplitude $A(t_0)$

and phase $\theta(t_0)$ depend on the initial time $t = t_0$. Then, the first-order equation is given by $d^2x_1/dt^2 + x_1 = -2dx_0/dt = 2A(t_0)\sin(t + \theta(t_0))$. Since the inhomogeneous term has the same angular velocity as the unperturbed one, the first-order solution necessarily contains a resonance term, and can be given in the following form

$$x_1(t; t_0) = -A(t_0)(t - t_0)\cos(t + \theta(t_0)),$$

which is made vanish at $t = t_0$ by adding an unperturbed solution, and hence $W_1(t_0)$ $= x_1(t_0; t_0) = 0$. This setting proposed in [48] is found to make the procedure of the RG method simple and transparent. In the present linear case, the first-order correction can be made vanish at the initial time, but when the equation has nonlinear terms, some terms would remain there. Similarly, the second-order solution can be given by $x_2(t; t_0) = (A(t_0)/2)(t - t_0)^2\cos(t + \theta(t_0)) + (A(t_0)/2)(t - t_0)\sin(t + \theta(t_0))$, which vanishes at $t = t_0$, and hence $W_2(t_0) = x_2(t_0; t_0) = 0$. Up to the third order of $\epsilon$, the perturbative solution is given by $x(t; t_0) = x_0(t; t_0) + \epsilon x_1(t; t_0) + \epsilon^2 x_2(t; t_0)$ with the initial value $W(t_0) = W_0(t_0)$. It is apparent that our approximate solution is diverging as $|t - t_0| \to \infty$ due to the secular terms, although it gives a sensible and hence valid solution locally around $t = t_0$. To obtain a valid solution in the global domain, we utilize the fact that the approximate solution should be valid around an arbitrary time $t = t_0$, and try to construct the envelope curve of the perturbative solutions, each contact point of which with the envelope should be chosen at $t = t_0$, such that the envelope function is given by the initial value $x_E(t) = x(t; t_0 = t) = W(t)$ $\forall t$. This can be accomplished through the following envelope equation, which is also identified with the RG equation:

$$\left.\frac{dx(t; t_0)}{dt_0}\right|_{t_0=t} = \left.\frac{\partial x(t; t_0)}{\partial t_0}\right|_{t_0=t} + \frac{dA(t_0)}{dt_0}\left.\frac{\partial x(t; t_0)}{\partial A}\right|_{t_0=t} + \frac{d\theta(t_0)}{dt_0}\left.\frac{\partial x(t; t_0)}{\partial \theta}\right|_{t_0=t} = 0. \quad (9)$$

This equation turns out to give the dynamical equations of the amplitude $A(t)$ and the initial phase $\theta(t)$. In fact, substituting the perturbation solution up to second-order to the RG Eq (9), we have

$$\left(\epsilon A(t) + \frac{dA(t)}{dt}\right)\cos(t + \theta(t)) + \left(-\frac{1}{2}\epsilon^2 A(t) - A(t)\frac{d\theta(t)}{dt}\right)\sin(t + \theta(t)) = 0. \quad (10)$$

To satisfy this condition for any time $t$, the amplitude $A(t)$ and the initial phase $\theta(t)$ should satisfy the following dynamical equations

$$\frac{dA(t)}{dt} = -\epsilon A(t), \qquad \frac{d\theta(t)}{dt} = -\frac{\epsilon^2}{2}, \quad (11)$$

respectively. These equations are readily solved to yield $A(t) = A_0 \exp(-\epsilon t)$ and $\theta(t) = -(\epsilon^2/2)t + \phi_0$, respectively. Inserting these expressions into the envelope function $x_E(t) = W(t) = x_0(t; t) =: x(t)$, we arrive at

$$x(t) = x(t; t_0)|_{t_0=t} = A_0 e^{-\epsilon t}\cos\left(\left(1 - \frac{\epsilon^2}{2}\right)t + \phi_0\right), \quad (12)$$

which is an approximate but globally valid solution to Eq (7): Indeed it has an infinite power series of $\epsilon$. It is also noteworthy that the angular velocity of Eq (12) gives a good

approximation of the exact one up to the third order, as $\omega = \sqrt{1-\epsilon^2} \approx 1 - \epsilon^2/2 + o(\epsilon^2)$. For more details with a nonlinear equations, see S1 Text.

### 3.3. RG analysis of waveform-period correlations

In Sects 2.1 and 3.1, we demonstrated that the temperature compensation of the period in the Goodwin model is accompanied by an increase of the index $NS$ and waveform distortion correspondingly occurs as temperatures increase. This raises the following question: Is there a universal law governing the waveform distortion occurring when temperature increases? To answer this question, we derive an approximate solution for the time evolution of the Goodwin model for the circadian rhythm using the renormalization group (RG) method [47]. The solution obtained using the RG method can be interpreted as the envelope of the set of solutions given in the perturbation theory, which has been applied to various models, including (but not limited to) ODE, PDE, discrete systems, and stochastic equations [48,49,52,53,58]. To apply the RG method, we again set the transcriptional regulation function $f(x_3)$ to be $r/x_3^n$. In this function, $n$ is the cooperativity of the transcriptional regulation, which is a Hopf bifurcation parameter. The approximate solution of the phosphorylated protein of the circadian clock reads (see S2 Text)

$$x_3(t) = \left(\frac{p_1 p_2 r}{s_3}\right)^{\frac{s_3}{s_1 s_2}} + \epsilon A_0 \sin(\omega t) + \epsilon^2 A_1 A_0^2 \sin(2\omega t + \alpha) + o(\epsilon^2) \quad (13)$$

where we have

$$\epsilon = n - \frac{s_4}{s_3}, \quad (14)$$

$$s_1 = k_1 + k_2 + k_3, \ s_2 = k_1 k_2 + k_2 k_3 + k_3 k_1, \quad (15)$$

$$s_3 = k_1 k_2 k_3, \ s_4 = (k_1 + k_2)(k_2 + k_3)(k_3 + k_1), \quad (16)$$

with the angular velocity and the phase parameter of the second-order term

$$\omega = \sqrt{s_2} - \epsilon \frac{s_1 s_3 s_4}{6(2 s_1 s_2^2 - (s_1^2 + 6 s_2) s_3)\sqrt{s_2}} + o(\epsilon^2), \quad (17)$$

$$\alpha = \arctan\left(\frac{s_1}{2\sqrt{s_2}}\right), \quad (18)$$

as well as the amplitudes

$$A_0 = \sqrt{\frac{4((s_1^2 + s_2)^2 - 2\epsilon s_1 s_3)(s_1^2 + 4 s_2) s_3^3}{\epsilon(2 s_1 s_2^2 - (s_1^2 + 6 s_2) s_3)(s_1^2 + s_2)^2 s_4 s_1 s_2}} \left(\frac{p_1 p_2 r}{s_3}\right)^{\frac{s_3}{s_1 s_2}}, \quad (19)$$

$$A_1 = \frac{s_4 s_1}{12 s_3 \sqrt{s_1^2 + 4 s_2}} \left(\frac{s_3}{p_1 p_2 r}\right)^{\frac{s_3}{s_1 s_2}}. \quad (20)$$

The RG method provides an approximate but globally valid solution, and thus enable us to make a detailed investigation of the waveform distortion when temperature compensation occurs in the Goodwin model. For example, we analyzed how the value of $A_1$ in Eq (20) depends on the reaction rates. This parameter is associated with the amplitude of the second harmonic component. To quantify its sensitivity to changes in the reaction rates, we used elasticity, defined as $\partial \ln A_1 / \partial \ln q_i$, where $q_i$ represents the parameters $k_1$, $k_2$, $k_3$, $p_1$, $p_2$, and $r$.

We found that $A_1$ increased or decreased with increasing degradation rates ($k_1$, $k_2$, and $k_3$), and consistently decreased with increasing phosphorylation and transcription rates ($p_1$, $p_2$, and $r$). The sum of the elasticities, $\sum \partial \ln A_1 / \partial \ln q_i$, was always zero, which reflects the fact that $A_1$ remains unchanged when all reaction rates are scaled by the same factor (S2 Fig).

The numerical analysis using the same parameter sets as used in Fig 2A, in which the relative period in the model remains stable and within the interval $(0.85, 1.0)$ against the temperature variations, shows that the phase parameter $\alpha$ in the 2nd-order frequency tends to decrease with increasing reaction rates (see Fig 3A). When the increase in reaction rates is small, the change in the phase of the second-order frequency scatters around zero and is negligible. However, with a significant increase in the reaction rates, the phase of the second order always tends to decrease as the reaction rates increase.

The significance of the phase parameter $\alpha$ given in the second-order term on the waveform of the time series can be understood intuitively as follows: when the phase $\alpha$ is large, the increasing duration tends to become longer because of the less overlap of the time profiles given by $\sin(\omega t)$ and $\sin(2\omega t + \alpha)$ (Fig 3B and 3C, blue line). Conversely, a smaller $\alpha$ tends to result in a shorter increasing duration because of an additive effect of the two terms, which leads to a steeper slope on total, as presented in Fig 3B and 3C, red line. Therefore, the numerical results in Fig 3A suggest that the decreasing duration of the time series elongates with as the reaction rate increases when the period is relatively stable despite the increasing reaction rate.

For a theoretical confirmation of the numerical result that the phase parameter $\alpha$ given in the second-order term tends to become smaller as the temperature increases when the period is temperature-compensated, we analyze the sensitivity of the angular frequency $\omega$ and the phase $\alpha$ to the reaction rates by utilizing the results of the RG method. With use

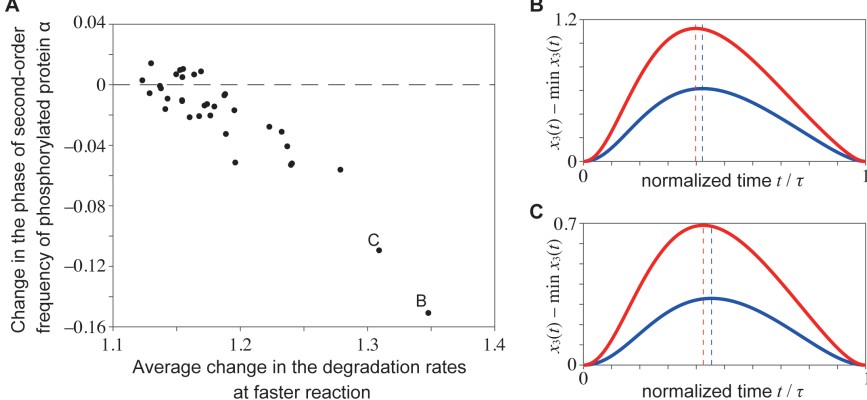

**Fig 3. Change in the phase of the second-order frequency $\alpha$ in phosphorylated protein oscillation under varying circadian clock model parameters.** (A) The parameters $k_1$, $k_2$, $k_3$, $p_1$, $p_2$, and $r$ were randomly increased from 1.1 to 1.9. The horizontal axis represents the arithmetic mean of the fast-case values of $k_1$, $k_2$, and $k_3$ divided by the slow-case values. The parameters $p_1$, $p_2$, and $r$ are omitted because they do not affect the period in Eq (17). The vertical axis presents changes in the phase of the second-order frequency obtained via generalized harmonic analysis of numerically calculated periodic time series. When the period length is maintained (relative period within 0.85-1.0), the phase of $\alpha$ of $x_3(t)$ tends to decrease as the reaction rate increases. (B, C) Time series for the increased reaction rates (1.1-1.9) while maintaining the period. The red and blue lines represent the time series for increased reaction rates and those for the reference parameter set, respectively. The parameter values are provided in S2 Table. For comparison, the period length is standardized to 1. The time of the minimum value is set to 0, and the maximum phosphorylated protein oscillation is indicated by the dotted line. Increased reaction rates tend to advance the timing of the maximum oscillation.

of Eqs (17) and (18), we calculate $d\omega = (\partial\omega/\partial k_1)dk_1 + (\partial\omega/\partial k_2)dk_2 + (\partial\omega/\partial k_3)dk_3$ and $d\alpha = (\partial\alpha/\partial k_1)dk_1 + (\partial\alpha/\partial k_2)dk_2 + (\partial\alpha/\partial k_3)dk_3$. In Fig 4A, we present the parameter regions given by the constraints $d\omega = 0$ (red surface) and $d\alpha = 0$ (yellow surface) for the cooperativity $n = 12$ of the transcription regulation. We can see that the region for $d\alpha < 0$ (outside yellow surface) includes that given by the constraint $d\omega = 0$ for all $k_i$ ($i = 1, 2, 3$). This implies that if the period is robust against a change in the parameters $k_i$ ($i = 1, 2, 3$), then the phase $\alpha$ in the second-order term always becomes smaller with increasing parameters.

Next, let us examine how the parameter regions given by the constraints $d\omega = 0$ and $d\alpha = 0$ change with variations of the cooperativity $n$. The numerical calculation shows that the region corresponding to $d\alpha < 0$ includes that given by $d\omega = 0$ for $n = 13$ and 14. However, in the case of an exceedingly high cooperativity of transcription regulation $n$, which is not biologically realistic, $d\alpha$ can be positive when $d\omega = 0$. For instance, for $n = 20$, although $d\alpha$ is negative for most of the parameter space, there *is* a region in which $d\alpha > 0$ when $d\omega = 0$ (see Fig 4B). These results indicate that if circadian rhythms are stable under temperature variations, the slope in the increasing phase of phosphoproteins should become sharper as temperature increases. Thus, we conclude that (i) the waveform of the gene activity rhythm should be more distorted at higher temperatures, and (ii) the rate of the increase in phosphoprotein levels should be greater at higher temperatures if temperature compensation is achieved. In principle, these features can be tested experimentally.

### 3.4. Verification of the theoretical analysis of temperature compensation using published experimental data

The period formula Eq (5) of the Goodwin model indicates that the non-sinusoidal index *NS* of the waveform of the circadian rhythms becomes larger, implying greater distortion of the waveform when all reactions are faster at higher temperatures during temperature compensation. To test this theoretical prediction of circadian gene activity in actual organisms, we analyze the waveform of the activity rhythms of the timeless gene in Drosophila at 18 and 29°C using published experimental data [59]. First, we extract the time series of the average

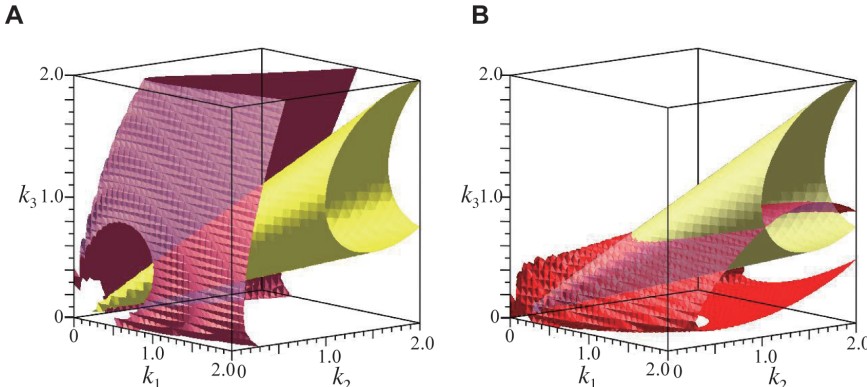

**Fig 4. The parameter space for the constant period ($d\omega = 0$, *red surface*) with increasing degradation rates $k_1$, $k_2$, and $k_3$, and that for the constant phase of second-order frequency ($d\alpha = 0$, *yellow surface*).** The phase of second-order frequency $\alpha$ should decrease with increasing $k_1$, $k_2$, and $k_3$ outside the yellow surface ($d\alpha < 0$). The cooperativity $n$ is $n = 12$ (A), and $n = 20$ (B), and the variations of the parameters are set to $dk_1 = 0.3$, $dk_2 = 0.2$, and $dk_3 = 0.1$. (A) When $n$ is small ($n = 12$), surface $d\omega = 0$ is included in space $d\alpha < 0$, which means that $\alpha$ should become smaller if $\omega$ does not change with increasing $k_1$, $k_2$, and $k_3$. (B) When $n$ is large ($n = 20$), surface $d\omega = 0$ can intersect $d\alpha = 0$ although high cooperativity is unrealistic.

curve from Fig 3C in a prior study [59]. Then, we add uniformly distributed noise between -0.4 and 0.4 to the extracted data to consider data errors (Fig 5A). The Fourier coefficients are estimated using generalized harmonic analysis (GHA) [32,68–70] (see Methods for details). The resultant *NS* of the activity rhythms of the timeless gene, as defined by Eq (1), at a higher temperature (29 °C) tend to be larger than that at a lower temperature (18 °C), whereas the *NS* values are somewhat varied (Fig 5B), which is consistent with the prediction.

Experimental studies have demonstrated that temperature compensation can be impaired by genetic mutations. In the Drosophila mutant perL, the period increases with increasing temperature [59,71]. Eq (5) implies that if the period increases with temperature, then the waveform of the circadian rhythm should be non-sinusoidal and more distorted at higher temperatures. Thus, it is predicted that the waveform of the circadian gene activity in perL should become more non-sinusoidal with higher temperatures. Again, we can quantify the waveform of perL using experimental data [59] (S3 Fig). The waveform of circadian gene activity tends to be more non-sinusoidal at higher temperatures in perL, as observed in the wild-type, whereas the *NS* values varied, in line with the prediction. To be precise, we performed a *t*-test to compare the two groups at higher and lower temperatures. The difference in estimated *NS* values between the two temperature conditions was statistically significant in the wild-type (*t* = 9.5755, *df* = 193, *p* < 0.001) (Fig 5). In the perL mutant, the difference was also statistically significant (*t* = 14.7971, *df* = 191, *p* < 0.001) (S3 Fig), supporting our theoretical predictions.

To examine the generality of the results, we also tested our conclusion using mouse embryonic fibroblasts (MEFs) data [21]. According to our prediction, the waveform of the circadian gene activity (Per2-luc) in wildtype mice should become more non-sinusoidal at higher temperatures. Indeed, the *NS* value was significantly higher at elevated temperatures (*t* = 2.9793, *df* = 189, *p* = 0.0016), supporting the generality of our findings (S4 Fig).

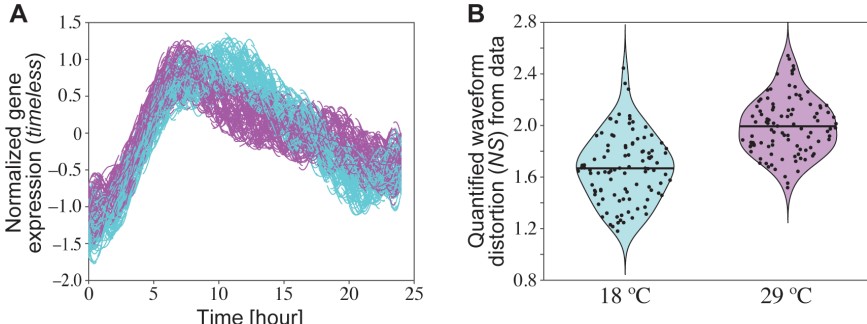

**Fig 5. Analysis of the circadian waveform of wild-type Drosophila at different temperatures using previously reported experimental data [59].** (A) Re-plot of Fig 3C in [59]. The average curves of *tim-luc* at 18 and 29 °C are extracted, and uniformly distributed noise between –0.4 and 0.4 is added to the extracted data, generating 100 time series datasets. Spline interpolation is applied to set the sampling interval to 0.1 h. The interpolated time series data at 18 (cyan) and 29 °C (magenta) are plotted. (B) Distribution of the quantified waveform distortion (*NS*) from the data at 18 (cyan) and 29 °C (magenta). The time series data with noise are detrended by multiplying an exponential function so that the positions of local minima of the oscillations are approximately equal. The Fourier coefficients of the detrended time series are quantified using GHA. The *NS* values are estimated from the coefficients up to the third harmonics. Dots represent the *NS* values for each dataset, and the horizontal lines represent their average values. The estimated *NS* values increased significantly at higher temperatures (*t* = 9.5755, *df* = 193, *p* < 0.001).

## 4. Theoretical analysis of synchronization in the circadian rhythm model

The numerical result in Fig 1B shows that the range of synchronization into the light-dark cycles tends to decrease as the waveform becomes more distorted in the simple circadian clock model. To clarify the condition for synchronization, we again consider the Goodwin model but with an external force incorporated as follows:

$$\frac{dx_1}{dt} = f(x_3) - k_1 x_1 + I\cos(\Omega t) \tag{21}$$

$$\frac{dx_2}{dt} = p_1 x_1 - k_2 x_2 \tag{22}$$

$$\frac{dx_3}{dt} = p_2 x_2 - k_3 x_3 \tag{23}$$

where $I\cos(\Omega t)$ is a periodic environmental change, such as a light-dark cycle. By eliminating $x_1$ and $x_2$, Eqs (21)-(23) is converted to the following single equation:

$$\frac{d^3 x_3}{dt^3} + s_1 \frac{d^2 x_3}{dt^2} + s_2 \frac{dx_3}{dt} + s_3 x_3 = p_1 p_2 f(x) + p_1 p_2 I\cos(\Omega t). \tag{24}$$

If the model is to admit a synchronization to the external cycle $I\cos(\Omega t)$ at all, then $x_3(t)$ should be written as the Fourier series $x_3(t) = \sum_{j=-\infty}^{\infty} a_j \exp(i\Omega j t)$. Multiplying Eq (24) by $dx/dt$ and integrating that for the interval $t$ to $t + 2\pi/\Omega$, we have the following equation:

$$\Omega^3 - \omega^2 \Omega = \frac{1}{2} p_1 p_2 I\mathcal{R}\sin\beta \tag{25}$$

where

$$\omega = 2\pi/\tau = \sqrt{s_2 \sum_{j=1}^{\infty} |a_j|^2 j^2 / \sum_{j=1}^{\infty} |a_j|^2 j^4}$$

is the natural angular frequency without the external force and

$$\mathcal{R} = \frac{|a_1|}{\sum_{j=1}^{\infty} |a_j|^2 j^4} \tag{26}$$

with $\beta$ being the argument of $a_1$ such that $a_1 = |a_1|\exp(i\beta)$. Because $-1 \leq \sin\beta \leq 1$, when $x(t)$ synchronizes with the external cycles, the angular frequency of the external cycles ($\Omega$) should satisfy the inequality

$$|\Omega^3 - \omega^2 \Omega| \leq \frac{1}{2} p_1 p_2 I\mathcal{R}. \tag{27}$$

We note that $\mathcal{R}$ defined by Eq (26) becomes smaller when the components of higher harmonics become larger and the waveform exhibits greater distortion. Therefore, if the waveform is more distorted by, say, an increasing temperature, the bounds of Eq (27) become smaller, and accordingly, the allowed region of the middle term is narrower. The left hand side of Eq (27) is a cubic function of $\Omega$, which is monotonically increasing near $\Omega = \omega$. If the waveform is more distorted, Eq (26) should be smaller, making the allowed region of Eq (27) narrower. Then, the range of $\Omega$ that causes synchronization becomes narrower, as presented in Fig 6. This indicates that the range of synchronization into light-dark cycles

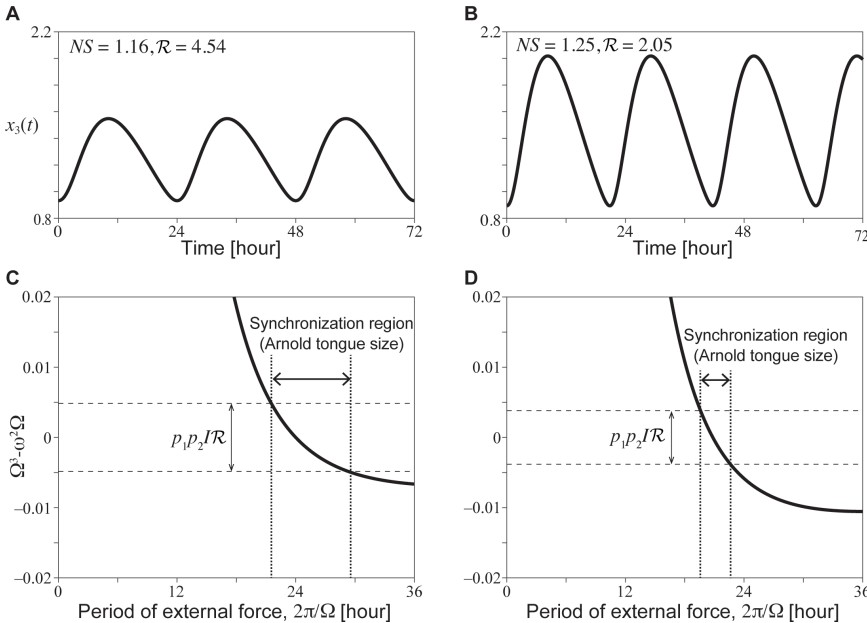

**Fig 6. The time series depicts cases in which the period length remains relatively constant despite increasing reaction rates in the circadian clock model (A: reference set case, B: increased reaction rates case).** Additionally, synchronization regions for these cases are presented (C: reference set case, D: increased reaction rates case). Panels A and B correspond to Fig 3B, but they include additional waveform values such as *NS* and $\mathcal{R}$. Synchronization regions for $I = 0.03$ in Eq (21) (C, D), calculated using Eq (27), are illustrated for each oscillation (A and B). Synchronization with the external forcing period occurs only when the cubic function $\Omega^3 - \omega^2\Omega$ falls within a specific range. This range becomes narrower as the waveform distortion increases along with reaction rates, consistent with the observations in Fig 1B.

always decreases as the waveform becomes more distorted in the simple circadian model, which is consistent with the numerical simulation in Fig 1B.

## 5. Discussion

We theoretically explored the conditions for clarifying the temperature compensation of the biological clock and its synchronization to light-dark cycles with a particular focus on waveform distortion. To investigate these conditions, we focus on the Goodwin model, which is widely used as a mathematical model that simulates various properties of biological clocks, including temperature compensation, synchronization to temperature cycles, and phase resetting by temperature steps [72]. The theoretical analysis of the Goodwin model revealed that waveform distortion of gene activity rhythms with increasing temperature is necessary for temperature compensation. Furthermore, we derived an approximate but globally valid solution to the waveform of the time profiles using the RG method as a powerful tool for global analysis. This allowed us to investigate, based on the analytical solution, whether there is a universal law for the mechanism by which the waveform changes with temperature variation. Notably, the relation between the period and waveform distortion holds not only for the Goodwin model but also for more realistic models. We have numerically showed the *NS* is larger when the period is stable to changes in temperature by using Zhou model, which includes 190 variables [21,32]. Additionally, we numerically analyzed the period and *NS* of Relógio model, which includes 19 variables [73], when all reaction rates increased by a factor of 1.5–2.5 (see S3 Text for details). When the period is stable or increases with increasing

reaction rates, geometric mean of *NS* becomes larger (S5 Fig). This suggests that the waveform distortion is important for temperature compensation in realistic circadian clock models.

The results indicated that temperature compensation is more likely to occur when the waveform is distorted, especially if the decreasing duration of circadian protein oscillation elongates as temperature increases. Although theoretical predictions based on a model might not always be realized in real organisms, we quantified the gene activity rhythms of published experimental data using Drosophila and mice. This quantification confirmed that the waveform is distorted at high temperatures, in accordance with our theoretical predictions.

It is notable that the systematic wave distortion governed by Eq (5), which we have found to hold in the Goodwin model, also applies to a wide class of non-linear oscillators used for biological phenomena different from biological rhythms, including the Lotka-Volterra model [74], which is commonly used in ecology, and the van der Pol model, as presented in the S4 Text and S5 Text, S6 Fig and S7 Fig. (see also [32]). This suggests that exploring the possible significance of waveform distortion in other mathematical models, such as the Fitz-Hugh-Nagumo model in neuroscience, would be intriguing [74–76].

To the best of our knowledge, this is the first study to apply the RG method, a powerful resummation method of the perturbation series first developed in physics, to circadian rhythm problems. In the RG method, secular terms appearing in the naïve perturbation series are renormalized into the 'integral constants' , which thus acquire the nature of the slow modes, making it a powerful tool for global and asymptotic analysis. Unlike the naive perturbation theory, the solutions given by the RG method provide a time evolution close to numerical simulations in the relevant global domain of time, offering an approximate solution for the period and waveform. The analytical results predict that longer tails of gene activity rhythms at higher temperatures occur for temperature compensation.

This study also investigated the synchronization with environmental light-dark cycles at various temperatures [35,36]. The numerical simulations and theoretical analysis predict that as the distortion of the gene activity rhythms for achieving a temperature-compensated period increases, it becomes more difficult to synchronize with the light-dark cycle. This prediction aligns with the reported temperature-dependent variation in response to light pulses in Drosophila and Neurospora, displaying smaller phase shifts at higher temperatures [60,61]. These results suggest that the waveform of gene and/or protein activity may vary seasonally between hot summers and cold winters, potentially affecting the entrainment of circadian clocks. Further studies are needed to investigate this possibility, particularly in light of seasonal variations in both temperature and light intensity.

To compare the role of waveform with that of amplitude in temperature compensation and synchronization, we also analyzed amplitude. Specifically, we numerically showed that the amplitude also becomes larger at higher temperatures when the period is stable to changes in temperatures (S1 Fig), which is consistent with previous study [22,31]. The correlation between period and amplitude is called "twist" [77]. It has been known that large amplitudes lead to narrow synchronization region [36,78]. These might indicate that both the large amplitudes and distorted waveform lead to similarly narrow synchronization region.

As mentioned in the Introduction, previous theoretical and experimental studies, such as those using Drosophila [59], suggested that waveforms in gene activity rhythms do not change under temperature variations, although they are blurred by error bands. These findings are apparently inconsistent with our current conclusion. We believe that this discrepancy stems from two main factors: differing assumptions about the temperature sensitivity of degradation rates and differing interpretations of experimental results regarding gene activity rhythms at distinct temperatures. First, the previous study assumed that all degradation rates are temperature-insensitive. Thus, the waveform of gene activity rhythms does not need to

change with temperature. By contrast, we assume that some degradation rates at least should accelerate with temperature, leading to a more distorted waveform of gene activity rhythms at higher temperatures. Second, the previous study [59] interpreted their experimental results as indicating that gene activity rhythms at different temperatures can be collapsed onto each other by rescaling, supporting their prediction that temperature compensation occurs because of rescaling and the temperature insensitivity of degradation rates. Conversely, our quantification of their experimental data indicated that the waveform tends to be more distorted at higher temperatures, whereas variation in *NS* values was noted. Thus, the present result is consistent with our theoretical prediction. Our theoretical prediction can be tested in circadian organisms such as mice and Neurospora [79]. We believe that further systematic quantification of the waveforms of gene activity and/or protein activity rhythms in various circadian organisms will be essential for clarifying the importance of the waveform in circadian rhythms in the future.

## Methods

### Computation of ODEs

The ordinary differential equations in this work were calculated by a fourth-order Runge-Kutta method with MATLAB (The MathWorks, Natick, MA). The time step size $\Delta t$ for the Goodwin model (Eqs (2)–(4)) was 0.001, that for the Lotka-Volterra model (Eqs (104)–(105)) and the van der Pol oscillator (Eq (114)) was 0.0001, and that for Relógio model [73] was 0.01.

### Numerical analysis of synchronization with light-dark cycles

The angular frequency of the light-dark cycle $\Omega$ in Eq (6) was changed by 0.0001. We considered that the model was synchronized with the light-dark cycle when the change in the amplitude was smaller than the rounding error as in Fig 1B.

### Estimation of waveform distortion from simulated time-series

We estimated waveform distortion, i.e., non-sinusoidal power (*NS*, see Eq (1)) from the simulated time-series by using the generalized harmonic analysis (GHA) method as in Figs 1B and 2A. The GHA method estimates the amplitudes $A_j$ and $B_j$, and the frequencies $f_j$, which minimize the squared residual $\int_0^L [x(t) - \sum_{j=1}^{j_{max}} \{A_j \cos(2\pi f_j t) + B_j \sin(2\pi f_j t)\}]^2 dt$. The detailed procedure of GHA can be found in [32,69].

### Estimation of phases of Fourier components from simulated time-series

By using GHA method, we computed the Fourier series $x(t) = \sum_{j=1}^{j_{max}} \{A_j \cos(2\pi f_j t) + B_j \sin(2\pi f_j t)\}$. This form was converted into $x(t) = \sum_{j=1}^{j_{max}} r_j \sin(2\pi f_j t + \alpha_j)$, where $r_j = \sqrt{A_j^2 + B_j^2}$ and $\alpha_j = \arctan(B_j/A_j)$.

### Computation of parameter space for temperature compensation using the RG method

The frequency change $d\omega = (\partial\omega/\partial k_1)dk_1 + (\partial\omega/\partial k_2)dk_2 + (\partial\omega/\partial k_3)dk_3$ and the phase change $d\alpha = (\partial\alpha/\partial k_1)dk_1 + (\partial\alpha/\partial k_2)dk_2 + (\partial\alpha/\partial k_3)dk_3$ were calculated from Eqs (17) and (18) by using Maple. The parameter space for $d\omega = 0$ and $d\alpha = 0$ in Fig 4A and 4B were numerically generated by Maple.

### Estimation of waveform distortion from published experimental data

The average curves of experimental time-series of *timeless* luciferase (*tim-luc*) reporter (shown to recapitulate the dynamics of *timeless* gene expression) at 18 and 29 °C of Figs 3C and 4B in the published literature [59] were extracted using WebPlotDigitizer at 1 h intervals. Uniformly distributed noise between –0.4 and 0.4 was added to the extracted data, generating 100 time series datasets so that the experimental noise is roughly reproduced as in Figs 5A and S3 FigA. Spline interpolation was applied to set the sampling interval to 0.1 h to smooth time series data. The time series data with noise were detrended by multiplying an exponential function so that the positions of local minima of the oscillations are approximately reproduced. The width of the window for the analysis is set to one period. The Fourier coefficients of the detrended time series were quantified using GHA. The *NS* values were estimated from the coefficients up to the third harmonics using Eq (1) with $m = 4$ and $q = 2$, as shown in Fig 5B and S3 FigB. To determine the significance of the differences, Welch's *t*-test was used.

## Supporting information

**S1 Text. Brief introduction of the renormalization group (RG) method using a simple model with Hopf bifurcation.**
(PDF)

**S2 Text. Derivation of a time-evolution solution in a circadian rhythm model using the RG method.**
(PDF)

**S3 Text. Numerical analyses of Relógio model.**
(PDF)

**S4 Text. Numerical and RG analyses of the Lotka-Volterra model.**
(PDF)

**S5 Text. Numerical and RG analyses of the van der Pol model.**
(PDF)

**S1 Fig. The period and amplitude of Goodwin model as function of temperature.**
(PDF)

**S2 Fig. Sensitivity of $A_1$ to the reaction rates in the Goodwin model.** We first generated 100 reference parameter sets yielding oscillations. Then the reaction rates of $k_1$, $k_2$, $k_3$, $p_1$, $p_2$, and $r$ were increased individually by 1%. We show the elasticity (= $\partial \ln A_1 / \partial \ln q_i$, where $q_i$ represents $k_1$, $k_2$, $k_3$, $p_1$, $p_2$, and $r$). The bar indicates the average, and the line indicates the standard deviation. The right column shows the summation of the elasticities (= $\sum \partial \ln A_1 / \partial \ln q_i$).
(PDF)

**S3 Fig. Analysis of the circadian waveform of the Drosophila mutant perL at different temperatures using previously reported experimental data [59].** (A) Re-plot of Fig 4B in [59]. The average curves of *tim-luc* at 18 and 29°C are extracted using WebPlotDigitizer at 1-h intervals. Noise uniformly distributed between –0.4 and 0.4 is added to generate 100 time series datasets. Spline interpolation is applied to set the sampling interval to 0.1 h. The interpolated time series data at 18 (cyan) and 29 °C (magenta) are plotted. (B) Distribution of the quantified waveform distortion (*NS*) from the data at 18 (cyan) and 29 °C (magenta). The noisy time series data are detrended by multiplying an exponential function to align local maxima at 18 °C or local minima at 29 °C. The Fourier coefficients of the detrended time-series are quantified using GHA. *NS* values were estimated from the coefficients up to the

third harmonics. Dots represent the *NS* values for each data set, and horizontal lines represent their average values. The estimated *NS* values increased significantly at higher temperatures ($t = 14.7971$, $df = 191$, $p < 0.001$).
(PDF)

**S4 Fig. Analysis of the circadian waveform of *Per2-luc* in mice at different temperatures using previously reported experimental data [21].** (A) Re-plot of Fig. 5C in [21]. The curves of *Per2-luc* between 5 and 30 hours at 30 and 37°C are extracted using WebPlotDigitizer at 1-h intervals. The amplitude is normalized so that the maximum value is one. Noise uniformly distributed between −0.05 and 0.05 is added to generate 100 time series datasets. Spline interpolation is applied to set the sampling interval to 0.1 h. The interpolated time series data at 30 (cyan) and 37 °C (magenta) are plotted. (B) Distribution of the quantified waveform distortion (*NS*) from the data at 30 (cyan) and 37 °C (magenta). The noisy time series data are detrended by multiplying an exponential function to align local maxima at 30 °C or local minima at 37 °C. The Fourier coefficients of the detrended time-series are quantified using GHA. *NS* values are estimated from the coefficients up to the third harmonics using Eq (1) with $m = 4$ and $q = 2$. Dots represent the *NS* values for each data set, and horizontal lines represent their average values. The estimated *NS* values increased significantly at higher temperatures ($t = 2.9793$, $df = 189$, $p = 0.0016$).
(PDF)

**S5 Fig. Distribution of the relative *NS* of the Relógio model [73] as a function of the relative period when rate constants are increased by a factor of 1.5–2.5.** We plot relative *NS* of cytoplasmic PER/CRY complex (A), *Cry* mRNA (B), and geometric mean of *NS* for all variables (C).
(PDF)

**S6 Fig. (A) Lotka-Volterra model. (B) Distribution of relative *NS* for $x(t)$ of the Lotka-Volterra model as a function of the relative period when rate constants are increased.** We first generated reference parameter sets. Then, each parameter $a$, $b$, $\varepsilon$, and $\varepsilon'$ in the model's reference parameter set was randomly multiplied by a factor of 1.1-1.9. (C) Examples of the waveform in the Lotka-Volterra model when the period is relatively unchanged. The blue and red lines represent small and large parameters, respectively.
(PDF)

**S7 Fig. The waveform examples of the van der Pol model.** The parameter value is $\varepsilon = 0.1$ (blue) and $\varepsilon = 3$ (red).
(PDF)

**S1 Table. Activation energies and frequency factors for each reaction in Fig 1.**
(PDF)

**S2 Table. Parameter values for each reaction in Figs 3B–3C and 6.**
(PDF)

## Acknowledgments

We thank H. Nakao, H. Chiba, Y. Kawahara, A. Mochizuki for useful comments on this study.

## Author contributions

**Conceptualization:** Shingo Gibo, Teiji Kunihiro, Tetsuo Hatsuda, Gen Kurosawa.

**Data curation:** Shingo Gibo.

**Formal analysis:** Shingo Gibo, Teiji Kunihiro, Tetsuo Hatsuda, Gen Kurosawa.

**Funding acquisition:** Teiji Kunihiro, Gen Kurosawa.

**Methodology:** Shingo Gibo, Teiji Kunihiro, Tetsuo Hatsuda, Gen Kurosawa.

**Project administration:** Gen Kurosawa.

**Supervision:** Gen Kurosawa.

**Validation:** Shingo Gibo.

**Visualization:** Shingo Gibo.

**Writing – original draft:** Shingo Gibo, Teiji Kunihiro, Tetsuo Hatsuda, Gen Kurosawa.

**Writing – review & editing:** Shingo Gibo, Teiji Kunihiro, Tetsuo Hatsuda, Gen Kurosawa.

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
