## [Decision Letter · Decision Letter 0]

3 Jun 2025

PCOMPBIOL-D-24-01483

Waveform distortion for temperature compensation and synchronization in circadian rhythms: An approach based on the renormalization group method

PLOS Computational Biology

Dear Dr. Gibo,

Thank you for submitting your manuscript to PLOS Computational Biology. After careful consideration, we feel that it has merit but does not fully meet PLOS Computational Biology's publication criteria as it currently stands. Therefore, we invite you to submit a revised version of the manuscript that addresses the points raised during the review process.

Please submit your revised manuscript within 30 days May 06 2025 11:59PM. If you will need more time than this to complete your revisions, please reply to this message or contact the journal office at ploscompbiol@plos.org. Please include the following items when submitting your revised manuscript:

We look forward to receiving your revised manuscript.

Kind regards,

Christian I. Hong, Ph.D.

Academic Editor

PLOS Computational Biology

Marc Birtwistle

Section Editor

PLOS Computational Biology

**Additional Editor Comments :**

The authors used the renormalization group method to show that waveform distortions are important for temperature compensation of circadian rhythms, which maintains a relatively stable period over a range of physiological temperatures. The reviewers agree that the described work addresses an important problem, temperature compensation, in the field of circadian rhythms, and that the authors have demonstrated convincing evidence supporting their conclusions. The reviewers also provided constructive comments ranging from the range of entrainment to readability and structure of the manuscript for a wider audience, which will strengthen the overall aspect of this work if addressed successfully.

**Journal Requirements:**

At this stage, the following Authors/Authors require contributions: Shingo Gibo, Teiji Kunihiro, Tetsuo Hatsuda, and Gen Kurosawa. Please ensure that the full contributions of each author are acknowledged in the "Add/Edit/Remove Authors" section of our submission form.

4) Please upload a copy of Supplementary Table 3 which you refer to in your text on page 11. Or, if the table is no longer to be included as part of the submission please remove all reference to it within the text.

5) We have noticed that you have uploaded Supporting Information files, but you have not included a list of legends. Please add a full list of legends for your Supporting Information files after the references list.

6) In the online submission form, you indicated that "All codes can be available upon requests."  All PLOS journals now require all data underlying the findings described in their manuscript to be freely available to other researchers, either

1. In a public repository

2. Within the manuscript itself

3. Uploaded as supplementary information.

7) Please amend your detailed Financial Disclosure statement. This is published with the article. It must therefore be completed in full sentences and contain the exact wording you wish to be published.

8) Please ensure that the funders and grant numbers match between the Financial Disclosure field and the Funding Information tab in your submission form. Note that the funders must be provided in the same order in both places as well. Currently, the order of the grants is different in both places.

**Reviewers' comments:**

Reviewer's Responses to Questions

**Comments to the Authors:**

**Please note that one of the reviews is uploaded as an attachment.**

Reviewer #1: This manuscript is an interesting follow-up of Ref.31 showing that the occurrence of temperature-compensation in biochemical oscillators is related to certain asymmetric concentration-time profiles. Here, the authors apply a renormalization method to show this. I have only a few comments, which I feel the authors should consider.

The suggestion that asymmetric concentration-time profiles appear necessary to get temperature-compensation seems to imply that when concentration profiles are completely symmetric, such as in feedbacks showing harmonic oscillations, then it should be difficult to introduce temperature compensation into these systems. Indeed, when analyzing a feedback mechanism that shows purely sinusoidal (i.e. harmonic) oscillations (see Fig S9 in Thorsen et al. PLoS ONE 9(9): e107766. doi:10.1371/journal.pone.0107766) this oscillator cannot be temperature compensated, because the frequency w there will depend only on the two rate parameters, k3 and VmaxEset, which both increase w according to w^2= k3 .VmaxEset. I feel that this could be used in this ms. as an additional indicator to the suggested rule addressed here.

Concerning the renormalization method: By going through the supporting material this method is quite theoretical/technical and requires some knowledge in perturbation theory. I think it would be good if the authors could include a section in the manuscript describing in ordinary terms the essence of this method with respect to oscillations for the less mathematically inclined reader (but still familiar with computations). Concerning Eq. 5: Although the authors refer to Forger’s paper, it would improve the ms. to briefly recapitulate the meaning of the aj’s and the method Forger has applied to obtain this equation. From the data by Thorsen et al (referred above) a similar analysis could probably be conducted on the expressions of the derived frequencies/periods there for various negative feedback oscillators, which included also a version of the Goodwin oscillator.

Line 54: The authors write: “One unresolved issue…..” and refer to several suggestions how temperature-compensation could be understood. I suggest to rephrase this sentence in the sense that these different approaches do not necessarily exclude each other and represent different aspects of how to approach/describe temperature-compensation. In his book “The Geometry of Biological Time” Art Winfree has a nice compilation of different suggested mechanisms for temperature-compensation (some not mentioned by the authors) to which the authors could/should refer to.

Also note that there is an earlier paper on the temperature-compensated Goodwin oscillator (JTB (1996) 179:275ff) describing many circadian clock properties, including temperature entrainment, phase resetting by temperature steps, etc, which the authors may refer to (since in this paper the authors focus on the temperature-compensated Goodwin oscillator).

Be aware that Goodwin formulated the oscillator using two versions: (i) one where removal reactions are zero-order with respect to the degraded species and where an inhibitory cooperativity of one is sufficient to obtain oscillations, and (ii) a second formulation where removal reactions are first-order with respect to the degraded species and which require a much higher cooperativity of the inhibitor. This has been summarized in a recent review by in Biotheoretica 2020, doi: 10.1007/s10441-020-09379-8, which may also be of interest to the reader with respect to the Goodwin oscillator.

Finally, concerning the verification of the model with respect of experimental data on page 13: the authors could further make connection to a paper from 1997 (Chronobiology International 14: 499-510, Fig. 3) which describes the temperature behavior of the per and frq mutants in Drosophila and Neurospora by the Goodwin oscillator.

There are also ultradian rhythms which show temperature-compensation. I feel the authors should extend their discussion briefly also in this direction.

Reviewer #2: The authors address a problem of general interest – the experimentally observed temperature compensation of circadian clock and the entrainment range of circadian rhythms by external zeitgebers.

Figure 1 summarizes the main findings based on numerical simulations of a specific model. The main achievements of the manuscript are the generalization of the findings regarding higher harmonics in the biological context (theories II, II, and III) and the generic discussion of the results using the renormalization group method.

I find the combination of extensive simulations (Figure 2), semi-analytical calculations (Figure 4), and experimental data (Figure 5) convincing. Other popular models (Lotka-Volterra, van der Pol, FitzHugh-Nagumo) confirm the main results.

Specific comments:

Line 27: Renormalization group theory was applied also earlier to biological problems. An example is the Feigenbaum scenario in iterated maps from population dynamics.

Line 35: Narrowing of the entrainment range is an important finding. However, also large amplitudes of strong oscillators lead to narrow entrainment ranges (see e.g. Aschoff and Pohl 1978). Is NS related to increasing amplitudes? Moreover, amplitude-period correlations termed “twist” lead to a skewing and possibly to a narrowing of Arnold tongues.

Line 89: I find the long list of references unfocussed. Moreover, classical old papers are not cited (Bogolubov, Wilson, Feigenbaum).

Line 209: Is “always” correct? I see some exceptions in Figure 2.

Line 272: There is some redundancy in Results and Methods (e.g. WebPlotDigitizer etc.).

Reviewer #3: see attachment please.

**Have the authors made all data and (if applicable) computational code underlying the findings in their manuscript fully available?**

Reviewer #1: Yes

Reviewer #2: Yes

Reviewer #3: None

PLOS authors have the option to publish the peer review history of their article (what does this mean?). If published, this will include your full peer review and any attached files.

Reviewer #1: **Yes: **Peter Ruoff

Reviewer #2: No

Reviewer #3: No

**Figure resubmission:**
---

## [Decision Letter · Decision Letter 1]

17 Jun 2025

Dear Dr. Gibo,

We are pleased to inform you that your manuscript 'Waveform distortion for temperature compensation and synchronization in circadian rhythms: An approach based on the renormalization group method' has been provisionally accepted for publication in PLOS Computational Biology.

Best regards,

Christian I. Hong, Ph.D.

Academic Editor

PLOS Computational Biology

Marc Birtwistle

Section Editor

PLOS Computational Biology

The authors have addressed all of reviewers’ previous concerns in the revised manuscript.

Reviewer's Responses to Questions

**Comments to the Authors:**

Reviewer #1: The authors have answered my comments/queries in a satisfactory manner.

Reviewer #3: The revisions required for this elegant study were carefully done by the authors.

**Have the authors made all data and (if applicable) computational code underlying the findings in their manuscript fully available?**

Reviewer #1: Yes

Reviewer #3: None

PLOS authors have the option to publish the peer review history of their article (what does this mean?). If published, this will include your full peer review and any attached files.

Reviewer #1: No

Reviewer #3: No

---

## [Editor Report · Acceptance letter]

PCOMPBIOL-D-24-01483R1

Waveform distortion for temperature compensation and synchronization in circadian rhythms: An approach based on the renormalization group method

Dear Dr Gibo,

I am pleased to inform you that your manuscript has been formally accepted for publication in PLOS Computational Biology. Your manuscript is now with our production department and you will be notified of the publication date in due course.

With kind regards,

Zsofia Freund
